# Quantification of Canine Apocrine Gland Anal Sac Adenocarcinoma (AGASACA) Tumor Specimen Shrinkage after Formalin Fixation

**DOI:** 10.3390/ani12151869

**Published:** 2022-07-22

**Authors:** Brandan G. Wustefeld-Janssens, Arathi Vinayak, Lindsay A. Parker, Danielle L. Hollenbeck

**Affiliations:** 1Flint Animal Cancer Center, College of Veterinary Medicine and Biomedical Sciences, Colorado State University, 300 W Drake, Fort Collins, CO 80523, USA; 2College of Veterinary Medicine and Biomedical Sciences, Texas A&M University, College Station, TX 77845, USA; lap177@mail.usask.ca (L.A.P.); dhollenbeck@cvm.tamu.edu (D.L.H.); 3Department of Surgery, VCA West Coast Specialty and Emergency Animal Hospital, Fountain Valley, CA 92708, USA; arathi.vinayak@vca.com

**Keywords:** apocrine gland anal sac adenocarcinoma, canine, formalin fixation, shrinkage

## Abstract

**Simple Summary:**

As the evidence in apocrine gland anal sac adenocarcinoma tumors grows, it is becoming more evident that stage-specific treatment strategies will become the mainstay. Thus, having confidence in the measurements of primary tumors is paramount. We aimed to quantify the degree of tumor tissue shrinkage after 24 and 48 h of formalin fixation to guide clinicians in their use of post-fixation measurements. We prospectively enrolled 23 client-owned dogs with naturally occurring apocrine gland anal sac adenocarcinoma that underwent surgical resection of at least the primary tumor. Measurements were recorded immediately before being placed in 10% buffered formalin and then again after 24 and 48 h of fixation, respectively. Overall, we found that tumors shrank by a mean of 4.8% and 7.2% after 24 and 48 h, respectively. This in real terms was a median of 1 mm. Other factors associated with the tumor, like the predominant microscopic pattern, the amount of necrosis, or the amount of fibrovascular stroma, did not have an impact on the degree of shrinkage. This study shows that the degree of shrinkage following formalin fixation should not impact the use of post-fixation measurements and can be used in clinical staging schema.

**Abstract:**

The aim was to prospectively measure the shrinkage of primary apocrine gland anal sac adenocarcinoma (AGASACA) tumors after 24 and 48 h of formalin fixation. Dogs that were diagnosed with AGASACA pre-operatively by aspiration cytology were prospectively enrolled in the study. Tumor extirpation was performed in a closed technique. The tumor and associated tissues were examined on the back table away from the patient and the widest dimension of the tumor was measured using a sterile ruler (Medline^®^; Northfield, IL, USA). This measurement was recorded in mm (t_0_). The tissue was placed in 10% buffered formalin and stored at room temperature. Two further measurements were taken after 24 (t_24_) and 48 (t_48_) hours of formalin fixation. Once the 48 h measurement was taken, the tissue was submitted for histopathology. The percentage of shrinkage between time points was calculated by using the following equation: (1 − [time b/time a]) × 100. Overall, 23 dogs with 23 tumors were enrolled. The mean percentage of shrinkage after 24 and 48 h of formalin fixation was 4.8% and 7.2%, respectively. The median diameter of the tumors reduced by 1 mm over 48 h and was not significantly different at any time point. These data will aid clinicians in interpreting measurements of AGASACA tumors following formalin fixation and shows that minimal change in tumor size is expected following 48 h.

## 1. Introduction

Apocrine gland anal sac adenocarcinoma (AGASACA) is a malignant epithelial neoplasm comprising 2% of all canine skin tumors and 17% of perianal malignancies, with the remaining tumor types consisting of perianal adenoma or adenocarcinoma, squamous cell carcinoma, and malignant melanoma [1,2,3,4,5,6,7,8]. A clinical staging scheme was proposed and then tested in a study of 130 dogs with naturally occurring diseases [5] (Table 1). In subsequent studies, this clinical staging scheme has been utilized and stage-specific outcomes have been reported [9,10]. In both studies cited, a recommendation can be provided for treatment modalities based on the clinical stage. For example, in the Meier et al. (2017) study, dogs with stage 3b disease had a median overall survival of 447 days for the group of dogs that had RT alone versus 182 days for surgery alone. In another study describing outcomes and prognostic factors in dogs with early-stage disease, dogs with primary tumors measuring less than 3.2 cm had a median survival of 1237 days when treated with surgery alone [10]. Since the scheme and studies are heavily dependent on the size of the primary tumor, the presence of metastatic disease to the regional lymph node(s), and the size of metastatic lymph node(s), it stands to reason that accurate measurement of both the primary tumor and the metastatic lymph node(s) is required. 

Tissue and tumor shrinkage after formalin fixation has been widely reported in human and veterinary medicine [11,12,13,14,15,16]. The change in the surrounding tissue is clinically relevant as the histologic tumor-free margin is highly predictive of recurrence in most canine solid tumors, and accounting for the degree of post-excision change in tissue dimensions may be important in decision making. This is especially important for highly elastic tissues like the intestine. In a study by Clarke et al. (2014), the mean length of intestinal specimens changed by an average of 26.3% after 24 h of formalin fixation. Additionally, having a precise tumor diameter for some tumor types, such as AGASACA, leads to a more accurate stage of disease and prognosis, resulting in a more tailored treatment plan.

A recent study comparing the accuracy of methods used to measure the primary tumor size in dogs with AGASACA found moderate agreement between formalin-fixed tissue, digital rectal examination, and computed tomography (CT) measurements [17]. Although the respective bias (mean difference) was small when digital rectal examination estimates were compared to CT measurements (0.15 cm), the 95% limits of agreement was relatively wide at −1.9 to 2.3 cm. This may be clinically significant where large errors may be presented especially in tumors measuring less than 4 cm. An interesting finding in the Schlag et al. (2020) study was that formalin-fixed tissue measurements more closely approximate the CT measurement when compared to digital rectal examination estimates. To the authors’ knowledge, the degree of AGASACA tumor shrinkage in formalin has not been studied in dogs. The objective of this study was to prospectively measure the shrinkage of primary AGASACA tumors after 24 and 48 h of formalin fixation. A secondary objective was to examine factors such as the make-up of the fibrovascular stroma, completeness of resection, and size of tumor on the degree of tissue shrinkage. 

## 2. Materials and Methods

Dogs that were diagnosed with AGASACA pre-operatively by aspiration cytology were prospectively enrolled in the study over two sites (Texas A&M University and VCA West Coast Specialty and Emergency Animal Hospital). Dogs were eligible for inclusion in the study if there was a palpable tumor associated with an anal sac and the owners consented to surgical resection via anal sacculectomy. Pre-operative cancer staging tests and pre-operative management were at the discretion of the attending clinician. In brief, the surgical procedure was performed under general anesthesia with the anesthetic protocol left to the discretion of the attending anesthesiologists. Tumor extirpation was performed in a closed technique and, in general, was as follows: a skin incision was created in a radial or parallel orientation relative to the fiber direction of the external anal sphincter muscle. The skin incision was performed either by a scalpel blade or by an electrosurgical monopolar handpiece, again dependent on the primary surgeon’s preference [18]. The subcutaneous tissues were divided, and a plane of excision was maintained at an arbitrary distance away from the tumor to ensure the tumor capsule was not breached. The anal sac duct was either ligated and divided with suture or was excised en bloc with a cuff of anal canal skin. The surgical site was lavaged with sterile saline (0.9% NaCl) and closed routinely. The tumor and associated tissues were examined on the back table away from the patient after the closure of the surgery site. The widest dimension of the tumor was measured using a sterile ruler (Medline^®^; Northfield, IL, USA) and this measurement was recorded in mm (t_0_). A sterile surgical marker was used to mark the line along which the widest diameter was measured to maintain consistency of where the tissue was measured at t_24_ and t_48_. The tissue was placed in 10% buffered formalin and stored at room temperature. Two further measurements were taken after 24 (t_24_) and 48 (t_48_) hours of formalin fixation. Personal protective equipment (disposable gown, gloves, and protective eyewear) was worn. The tissue was removed from formalin under a vent hood, gently blotted with a disposable paper towel to wipe away excess formalin, and measured at the widest point. Once the 48 h measurement was taken, the tissue was submitted for histopathology. Depending on the size of the samples, the tissue was either trimmed or not. At least one section of the tumor and, if appropriate, sections of the clinical margins were placed in histology cassettes, embedded in paraffin, cut as 4 µm sections, and stained with hematoxylin and eosin using standard procedures. The histological sections were evaluated by a board-certified pathologist. All histopathology reports were reviewed to confirm AGASACA diagnosis, and the following characteristics were recorded: the description of fibrovascular stroma, degree of necrosis, mitotic count, and the completeness of the histologic margin using the traditional residual tumor (R) classification system [19]. Tumors were assigned R0 if there were negative histologic margins and R1 if tumor cells extended to the cut edge or positive margin. The description of the stroma was recorded with the descriptors fine, moderate, and dense. Necrosis was recorded as none, rare, mild, moderate, and cystic. Mitotic count was defined as the number of mitotic figures in a field area of 2.37 mm^2^. Tumors were classified based on a published scheme into T-stage 1 (<2.5cm) or T-stage 2 (≥2.5 cm) [5]. 

### Data Analysis

Data were tabulated and continuous data were tested for normal distribution using the Shapiro–Wilk test. The data were expressed as median (95% confidence interval (CI)) if not normally distributed (age, body weight, measurements at t_0_, t_24_, and t_48_, and mitotic count) and comparisons were drawn between groups with the Mann–Whitney U-test. Kruskal–Wallis tests were used for multiple comparisons across groups not normally distributed. Categorial data were expressed as frequency (%). The percentage of shrinkage between time points was calculated by using the following equation: (1 − [time b/time a]) × 100 where time a was the first of the time points and time b was the latter [15]. Mean percentage shrinkage was calculated. The change in diameter was expressed as a negative value if the tumor shrank or positive if the tumor increased in size. A *p*-value of less than or equal to 0.05 was considered statistically significant. 

## 3. Results

Overall, 23 dogs with 23 tumors enrolled in the study between 1 February 2019 and 1 November 2020. Ten cases were enrolled at Texas A&M University and 13 cases at VCA West Coast Specialty and Emergency Animal Hospital. There were ten female dogs (nine neutered and one intact) and 13 male dogs (all neutered). Eighteen dog breeds were represented with the most common being cross-bred dogs (n = 5; 22%) and Golden Retrievers (n = 3; 13%). The median body weight was 23.8 kg (95% CI: 10.6–28.6) and the mean age was 11.1 +/− 2.6 years. 

The tumors were left-sided in 39% of cases and right-sided in 61%. The median mitotic count was 12 (95% CI: 9–22) with a range of 3 to 234. Tumor necrosis and the fibrovascular stroma were commented on in all 23 histopathology reports. Necrosis was absent in 13 (57%), rare in two (9%), mild in three (13%), moderate in two (8%), and cystic in three (13%). The fibrovascular stroma was described as fine in four (17%), moderate in ten (44%), and dense in nine (39%). The histologic margin was classified as R0 in 15 tumors (65%) and R1 in the remaining eight tumors (35%). The T-stage was classified as T1 in 14 tumors (61%) and T2 in nine tumors (39%). 

The median tumor size and median difference between time points are represented in Table 2. The median size of the tumor before fixation was 15 mm (95% CI: 16.6–33.9) with a range of 6 to 73 mm. Overall the median shrinkage over 48 h of formalin fixation was 1 mm with a range of 1 mm increase in diameter to 6 mm reduction. The mean percentage difference in size between T_0_ vs. T_24_ was −4.8% and T_0_ vs. T_48_ was −7.2%. The median diameter of tumors between T_0_ vs. T_24_ (*p* = 0.74) and T_0_ vs. T_48_ (*p* = 0.68) were not significantly different. The mean percent shrinkage between tumors with an R0 vs. R1 margin and between T-stage 1 vs. 2 tumors are represented in Table 3. There was no significant difference between the median diameter of tumors at any time point between those that were resected with an R0 or R1 margin (T_24_
*p*-value = 0.54; T_48_
*p*-value = 0.69) or between T-stage 1 or 2 tumors (T_24_
*p*-value = 0.18; T_48_
*p*-value = 0.48) (Table 2). The thickness of fibrovascular stroma did not influence the amount of tumor shrinkage at any time point between groups (*p* = 0.30). 

## 4. Discussion

The primary aim of the study was to measure the degree of shrinkage of AGASACA tumors after 24 and 48 h of formalin fixation. The percent decrease in size from immediately after removal to 24 and 48 h was 4.8 and 7.2%, respectively. No tumor factors were found that influenced shrinkage in a significant manner. 

There is a distinction to be considered between tumor and surrounding tissue shrinkage as both contribute to the overall specimen size but also individually can be used in different clinical scenarios. For example, in tumor types where tumor size is an independent predictive factor for the outcome, such as oral malignant melanoma and mammary tumors, this information can be used to predict biologic behavior and guide decisions [20,21,22]. Thus, knowing to what degree tumors shrink in formalin fixation may be valuable so that measurements taken after fixation can be interpreted in this context. The shrinkage noted in this study was not out of the range that has been previously reported. In a study measuring the shrinkage of cutaneous mast cell tumors and the surrounding tissue, the investigators found that overall specimen (tumor + surrounding normal tissue) shrinkage was 17% but the mean tumor-specific shrinkage was 4.5% [15]. Similarly, in a study of human esophageal carcinoma specimens, the authors found that the tumor shrunk on average 10% compared to approximately 50% of the total specimen length [23]. 

During the current study, there was no attempt to quantify the amount of peritumoral connective tissue which could significantly impact the amount of specimen shrinkage, although great care was taken to ensure that the tumor itself was measured. The surgical dose is heterogenous in AGASACA tumor resections, which is dictated mostly by anatomy (e.g., proximity to the anus and rectum) and tumor size whereby larger tumors have relatively less surrounding tissue. Although not entirely applicable, the standard of care surgical dose would be classified as marginal using the Enneking definition, which is defined as the plane of dissection being through the tumor pseudocapsule or in the tumor-reactive zone [24]. Typically, this means the volume of peritumoral tissue should be minimal and would have little impact on the change in volume of the overall specimen following formalin fixation. This, though, is a significant factor in other tumor types like canine cutaneous mast cell tumor and soft tissue sarcomas, where the standard of care for resection is wide and the histologic tumor-free margin is predictive of outcome [25,26]. In this case, accounting for the amount of connective tissue shrinkage around a tumor can then be utilized to calculate the optimal amount of unfixed tissue required to ensure a safety zone of resection and in theory would have more bearing on the overall local control rate [15]. This is not so in AGASACA tumors where the extent of resection is always limited by anatomy and no study to date has proven that histologic margin is predictive of outcome [27]. Nevertheless, as a surrogate indicator, we analyzed if there was a difference between those tumors that were resected with a complete margin (R0) versus those that were resected incompletely (R1). We assumed that there was some tissue surrounding the tumor in completely resected tumors and thus may have a larger degree of shrinkage. This logic has limitations, namely a complete resection does not infer the quantity of tissue, and a tumor with a 1 cm cuff of tissue would be grouped with a tumor with a 1 mm cuff. Secondly, an incomplete resection does not necessarily imply that there was no peritumoral tissue in all planes. There was no difference in the median shrinkage of tumors that were resected with a complete margin compared to an incomplete margin, but this result should be interpreted with caution and with the above in mind.

We evaluated histologic features that may have an impact on tumor shrinkage such as tumor necrosis, predominant histologic pattern, and the amount of fibrovascular stroma present. These variables were selected as they are commonly described in histologic reports and may, depending on the degree, account for a variation in shrinkage. The authors are unaware of other studies of AGASACA in dogs where any variable has been associated with tumor size and therefore theoretically inversely with the degree of shrinkage. The data herein would suggest that there is no impact on these variables on shrinkage post formalin fixation. 

Variation in measurement is a potential source of error. This study was conducted over two sites with multiple clinicians performing the measurements. The methodology of measuring tissue shrinkage in the Upchurch et al. (2018) study was described in a study of basal cell carcinoma in people [28]. Briefly, the edge of the tumor was marked by incising the dermis at the palpable edge of the tumor and marked with tissue ink. These defined, easily identifiable points were then used as reference points for subsequent measurements. This allowed both overall specimen shrinkage and tumor tissue-specific shrinkage to be assessed. This approach was not feasible in our study because AGASACA tumors are subcutaneous and a clear margin between the normal tissue and tumor edge is often not distinct, particularly after formalin fixation. We used the widest dimension instead and marked a line along the widest diameter for consistency. This was repeated over the two sites but measuring error was still possible. We ensured that the same observer performed all the measurements for each individual tumor to minimize error. We did not have multiple observers for each tumor, which could have been interesting to calculate interobserver variation or at least try to quantify the variation that was incorporated into the data. An additional step that was not taken in the above study would have been to incorporate high-definition photographs so that multiple observers could have verified measurements and ensured that they were accurately recorded for data analysis. 

Some tumors increased in size during fixation, which had been previously reported in the veterinary [15,29] and human literature [30]. The reasons for this are unknown and have been attributed to an unknown interaction between the tumor specimen and formalin. In the Kerns et al. (2008) study, specimens shrank in length and width immediately after excision and then on average, re-expanded a small amount after formalin fixation. This parallels a study in veterinary medicine examining the degree of shrinkage in post-excision and post-fixation small intestinal specimens [12]. The degree of shrinkage immediately post-excision was dramatic at a mean of 28%, but after 24 h of formalin fixation, the specimens on average increased in length by 3%. Granted, the degree of elastic connective tissue in small intestinal specimens is considerably more than in AGASACA tumor specimens, but the phenomenon of a slight increase in the size of specimens has been repeatedly reported. The specimens in the Clarke et al. (2014) study were normal, adding an extra layer to the discussion. 

## 5. Conclusions

There was a mean tumor shrinkage of 7.2% after 48 h of formalin fixation. No tumor-related factors that were assessed had a significant impact on the degree of shrinkage. These data will aid clinicians in interpreting measurements of AGASACA tumors following formalin fixation, will aid in incorporating post-fixation measurements into staging schema, and show that minimal change in tumor size is expected following 48 h. 

## Figures and Tables

**Table 1 animals-12-01869-t001:** Clinical staging scheme for dogs with AGASACA.

Stage	
I	T < 2.5 cm N0 M0
II	T > 2.5 cm N0 M0
IIIa	T any N1 (<4.5cm) M0
IIIb	T any N1 (>4.5cm) M0
IV	T any N any M1

**Table 2 animals-12-01869-t002:** Median tumor size and absolute shrinkage between time points.

	Median	Range (mm)		95% CI	
		Minimum	Maximum	Lower	Upper
T_0_ (mm)	15.0	6.0	73.0	12.0	30.0
T_24_ (mm)	14.0	6.0	73.0	11.0	26.0
T_48_ (mm)	15.0	5.0	73.0	10.0	28.0
T_24_-T_0_ (mm)	−1.0	−6.0	0.0	−1.0	0.0
T_48_-T_0_ (mm)	−1.0	−6.0	1.0	−2.0	−1.0

**Table 3 animals-12-01869-t003:** Shrinkage of tumors as mean % at different time points, between R-classification and T-stage.

Parameter (as % Change)	Time Points					
	T_24_-T_0_			T_48_-T_0_		
	Mean	Range		Mean	Range	
		Min	Max		Min	Max
Overall	−4.8	0	−20.0	−7.2	7.1	−25.0
R0	−5.0	0	−14.3	−7.8	7.1	−25.0
R1	−4.3	0	−20.0	−6.3	0	−16.7
T-Stage 1	−5.1	0	−14.3	−9.4	7.14	−25.0
T-Stage 2	−4.3	0	−20.0	−4.0	0	−9.7

## Data Availability

Data available on request.

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
