# Peer review of "Quantification of Canine Apocrine Gland Anal Sac Adenocarcinoma (AGASACA) Tumor Specimen Shrinkage after Formalin Fixation"

_animals, 2022, doi:10.3390/ani12151869_

Round 1
Reviewer 1 Report
In the discussion of Table 2, it says that the median diameters of tumors between T0 and T24 were and between T0 and T48 were not significantly different. This is confusing because the confidence intervals presented in Table 2 do not contain zero. So is this discrepancy because you used a nonparametric test (whereas the CI is presumably parametric) for the p-values, or did you not use a paired test here, or something else? Usually, one can invert a confidence interval into an hypothesis test and so the two should agree.
Author Response
You are correct, thank you for noticing this. The 95% CI have been corrected and are now of the medians. The p-values are correct
Reviewer 2 Report
The article describes the tumor specimen shrinkage in apocrine gland anal sac adenocarcinoma. It is a paper with a very simple methodology but a very interesting idea. For the Veterinary Oncology field, this research is really interesting and useful in clinical practice. The methodology is well presented and the manuscript add to current literature a very interesting result. Since I have some background in pathology, and authors performed a prospective study, I really felt a lack of images of different comparisons. Since authors have the measurement in different times, images comparing fresh samples and fixed after 24 and 48 hours, it will be interesting. It will be also very interesting some histological images, since authors provided some information regarding tumor characteristics, such as mitosis.
Overall, it is a very well designed and interesting research and I congratulate the authors for performing such a nice research.
Author Response
Thank-you for your review. We do not have pictures of the gross specimens or histology but this is something we could get and include.
Reviewer 3 Report
In this work the authors measure shrinkage of primary AGASACA tumors after formalin fixation and evaluate possible factor conditioning it.
Since many prognostic studies have emphasized the value of tumor size as well as measurement of the excision margins, the topic is of interest for both veterinary oncologists and pathologists.
The manuscript is clear and presented in a well-structured manner and I have very few comments about it.
Introduction: Is "Goldschmidt et al. Skin tumors of the dog and cat, Pergamon Press 1992" the most recent available reference about the incidence of AGASACA? Could you please check if some more updated data are available?
Material and Methods: could you please explain in more detail how the trimming was performed in order to ensure the best evaluation of histologic margins and degree of mitosis.
Could you please report the mitotic count reffered to a field area of 2,37mm2 instead of 10 high power fields as suggested by literature to standardize the evaluation of mitotic count
Results: please could you check the number of dogs enrolled into the study, You first report 23 dogs and subsequently you describe 10 female and 14 male dogs enrolled.
Author Response
Introduction: Is "Goldschmidt et al. Skin tumors of the dog and cat, Pergamon Press 1992" the most recent available reference about the incidence of AGASACA? Could you please check if some more updated data are available?
These incidence numbers reported in this reference are the ones that are repeatedly quoted in studies since the 1992 publications. To our knowledge there are updated incidence statistics reported for this disease.
Material and Methods: could you please explain in more detail how the trimming was performed in order to ensure the best evaluation of histologic margins and degree of mitosis.
Details on processing has been added to the material and methods between lines 93-98
Could you please report the mitotic count reffered to a field area of 2,37mm2 instead of 10 high power fields as suggested by literature to standardize the evaluation of mitotic count
Your point is well taken and this has been changed in the manuscript
Results: please could you check the number of dogs enrolled into the study, You first report 23 dogs and subsequently you describe 10 female and 14 male dogs enrolled.
Thank you for noticing this. There were 23 dogs. This has been corrected in the results